# Genetic Variability in the Physicochemical Characteristics of Cultivated *Coffea canephora* Genotypes

**DOI:** 10.3390/plants13192780

**Published:** 2024-10-04

**Authors:** Hilton Lopes Junior, Rodrigo Barros Rocha, Alana Mara Kolln, Ramiciely Nunes de Paula Silva, Enrique Anastácio Alves, Alexsandro Lara Teixeira, Marcelo Curitiba Espíndula

**Affiliations:** 1Legal Amazon Biodiversity and Biotechnology Network (Bionorte), Federal University of Rondônia (UNIR), Porto Velho 76801-058, RO, Brazil; 2Federal Institute of Education, Science and Technology of Rondônia (IFRO), Jaru 76890-000, RO, Brazil; alana.kolln@ifro.edu.br (A.M.K.); ramicielynunesdepaula@gmail.com (R.N.d.P.S.); 3Brazilian Agricultural Research Corporation (EMBRAPA Coffea), Brasília 70770-901, DF, Brazil; rodrigo.rocha@embrapa.br (R.B.R.); alexsandro.teixeira@embrapa.br (A.L.T.); marcelo.espindula@embrapa.br (M.C.E.); 4Capixaba Institute for Research, Technical Assistance and Rural Extension (INCAPER), Vitória 29052-010, ES, Brazil; 5Brazilian Agricultural Research Corporation (EMBRAPA Rondônia), Porto Velho 76801-058, RO, Brazil; enrique.alves@embrapa.br

**Keywords:** Rondônia coffee, chemical quality, selection gains, genetical diversity

## Abstract

The objective of this study was to characterize the genetic divergence and selection gains of the physicochemical grains traits of 68 genotypes of *C. canephora* most cultivated in the Western Amazon. For this purpose, the following characteristics were evaluated over two harvests: aqueous extract, ash, acidity, pH, protein, ether extract, soluble solids, phenolic compounds, soluble sugars, reducing sugars, and non-reducing sugars. The genotype × measurement interaction effect was significant for all characteristics, with a predominant simple interaction, resulting in smaller changes in the ranking of genotypes. Out of a total of 45 genotypic correlation estimates, 8 were significant, of which 5 were related to acidity. The dispersion of the first two components associated with reference points shows that the genotypes BRS3193, AS1, AS2, AS3, N16, CA1, and AS7 were closest to the ideal type of higher performance. Selection for the main characteristic of soluble sugars resulted in estimates of genetic progress lower than those observed using selection indices. The genetic materials present high genetic diversity, allowing the selection of reference plants with high levels of sugars (BRS3193, AS3, GJ25, and LB30), proteins (BRS2357), lipids (GJ30), and phenolic compounds in their green beans (BRS3193) and high water solubility (AS2).

## 1. Introduction

The cultivation of coffee in the Western Amazon has undergone significant technological expansion. Over the past 10 years, there has been a decrease in the planted area alongside a significant increase in productivity, due to the selection of plants and improvements in management practices [1].

The genetic material cultivated in the state of Rondônia consists of intervarietal hybrids with characteristics of the Conilon and Robusta botanical varieties. Conilon seeds were introduced by migrants from Espírito Santo, while the Robusta seeds were distributed by Embrapa in the 80s. In collaboration with the Agronomic Institute of Campinas (IAC), Embrapa introduced several progenies of the Robusta variety [1].

In the 90s, cloning techniques were introduced, transforming the coffee industry by allowing the cultivation of plants with the best agronomic traits [2]. Genotypes with hybrid characteristics between Conilon and Robusta, either empirically selected by coffee growers or developed by Embrapa, form the basis of coffee cultivation in Rondônia. These genotypes are known for combining the Robusta’s beverage quality, with fruity and chocolaty notes, or less pronounced nuances of Conilon [3,4,5].

The knowledge of the genetic variability, as studied in this work, challenges the long-standing paradigm that the *C. canephora* is associated with lower quality and market value. The beverage from this species was labeled as neutral, flavorless, and intended solely for the production of instant coffee or blends with Arabica coffee [6]. This recognition reflects the growing appreciation of the species, which is increasingly consumed by the population. In addition to having high productivity and adaptability in tropical climates [7], it also has chemical characteristics that tend to increase commercial interest [8].

One of the most important chemical properties is the soluble solids content. High levels of soluble solids tend to increase water solubility, enhancing the body of the beverage and being of significant interest to the instant coffee industry. The soluble solids fraction primarily consists of sugars, caffeine, trigonelline, and chlorogenic acids [9], which are naturally higher in *C. canephora* coffee plants [10].

Often, the sensory characteristics are associated with the concentration of chemical compounds present in its beans. High acidity, when combined with a profile of sweetness, bitterness, and aroma, tends to result in a beverage of superior quality [11].

The aim of this study was to characterize the genetic divergence and selection gains of the physicochemical properties of coffee beans from 68 *C. canephora* genotypes that are most widely cultivated in the Western Amazon.

## 2. Results and Discussion

Coffee cultivation in the Western Amazon region takes place in Am and Aw climates, which are characterized as tropical, warm, and humid [12]. The total annual precipitation for the 2020/2021 crop year, spanning from July 2020 to July 2021, was 2.112 mm, while for the 2021/2022 crop year (July 2021 to July 2022), it was 1.847 mm (Figure 1).

It is observed that the first measurement recorded higher annual accumulated precipitation, and with the exception of April 2022, the first crop also showed higher monthly accumulated precipitation (Figure 1). It is known that water deficits in coffee plants can lead to flower drop and damage to fruit growth, resulting in reduced bean filling [13,14,15]. Comparing the overall averages of the two measurements (Table 1), it is noted that the parameters Total Titratable Acidity (TTA), Total Crude Protein (TCP), Total Soluble Solids (TSS), ratio, Soluble Sugars (SS), and Total Reducing Sugars (TRS) showed significant differences between the measurements, which may be influenced by environmental factors. However, Aqueous Extract (AE), Total Ash (TA), Hydrogen Potential (pH), Ether Extract (EE), Total Phenolic Compounds (TPC), and Non-Reducing Sugars (NRS) did not show significant differences between the measurements.

The average concentrations of the variables TCP, TSS, ratio, SS, and TRS were higher in the first measurement, where the annual accumulated precipitation was greater. In contrast, the average TTA content was higher in the second measurement (Table 1).

Genotype × Years (G × Y) interactions were significant for all physicochemical traits, indicating the presence of genotypes with different performance between measurements (Table 1). The analysis of repeated measures over time allows us to consider temporary and permanent effects of experimental error. The genotype × years interaction was classified as simple (Figure 2), considering that changes in performance resulted in only small changes in the genotypic classification.

Repeatability, also referred to as the upper limit of heritability, is estimated by considering both its permanent and temporary components. Since this estimate takes into account not only additive genetic variance but also environmental factors that consistently impact the individual over time. Except for pH, which had a low repeatability estimate (r = 34.91), the other traits had repeatability estimates ranging from 55.93 to 74.42 (Table 1), which can be interpreted as moderate to high magnitudes [16]. The traits can be ranked based on their repeatability estimates as follows: TCP > AE ≥ EE > TPC > TA > TTA > SS ≥ NRS > TRS > TSS > ratio > pH.

The experimental coefficient of variation (CVe), estimated based on the average performance of the traits and the experimental error estimate, was assessed to gauge the precision of the experiments. All traits showed low CVe estimates, indicating good experimental precision, with values ranging from 0.37% for non-reducing sugars to 6.41% for pH (Table 1). Ratios greater than one between the genotypic coefficient of variation (CVg) and the coefficient of variation (CVe) indicate genetic progress through plant selection [17]. This ratio ranged from 0.92 for TSS to 4.81 for TPC. Based on this analysis, the traits can be ranked as follows: TPC > AE > TA > TCP > TRS > SS > NRS > EE > TTA > pH > Ratio > TSS.

The clustering of genotypes using Scott-Knott’s mean test indicates that some traits, such as Total Phenolic Compounds (TPC), exhibited high variability, forming 17 groups, while others, such as Total Soluble Solids (TSS), showed less variability, forming 4 groups. In our study, the levels of soluble sugars, proteins, and phenolic compounds varied from to 3.66 to 9,84%, 11.63 to 18.93%, and 4.2 to 5.86 g of gallic acid equivalent per 100 g, respectively (Appendix A). Reducing and non-reducing sugars varied from 0.86 to 1.97% and 2.23 to 8.05%, respectively. Among the non-reducing sugars, sucrose is the primary constituent of this class [18], and higher levels of soluble sugars positively influence sucrose concentration.

The aqueous extract (AE) reflects the content of substances in coffee beans that are soluble in boiling water, while Total Soluble Solids (TSS) represent the compounds soluble in water at room temperature [19]. These traits showed substantial genetic variability, with some clones, such as AS2, having high levels, while others, including GJ8, N2, and GB7, exhibited much lower levels. The WE content ranged from 27.38 to 33.21%, and the TSS ranged from 28.97 to 35.88% (Appendix A).

The relationship between the soluble fraction, specifically total soluble solids and total titratable acidity, is associated with the perception of sweetness. When this ratio is unbalanced, it can create a sensation of the product being “diluted” or “too acidic”. In our study, approximately 70% of the genotypes had a ratio exceeding 0.2. This finding aligns with a previous study [20], which noted that these range are not related with intrinsic sweetness perception. However this trait is associated with low acidity and high levels of soluble compounds found in the beans of this species. In this study, the acidity of the green beans ranged from 140.23 mL to 184.33 (NaOH 0.1 mol·L^−1^ em 100 g sample). The pH, which tends to be influenced by the acidity of the beans, ranged from 5.06 to 5.32 (Appendix A).

The ether extract (EE) levels ranged from 3.45 to 7.89%, with some genotypes, such as GJ30, exhibiting high concentrations (7.89%), and others, such as BRS2357, showing lower concentrations (3.45%) (Appendix A). Reported EE levels ranging from 3.76 to 6.48% within the same species [21]. The observed EE content was higher than at 10.90%. Other study that compared to Apoatã, Bukobensis, Laurentii, Guarani, and Conilon, observed that Conilon genotypes had lower values, close to 7.30% [22].

Total ash content ranged from 4.18 to 5.47% (Appendix A). The presence of nitrogen in the ash of *C. canephora* is usually higher than that of other nutrients, followed by potassium, calcium, phosphorus, sulfur, magnesium, iron, boron, manganese, copper, and zinc [23].

Analyzed the physicochemical characteristics of seeds from *C. canephora* genotypes of the Apoatã variety. Their study found an average ash content of 4.07%, ranging from 3.45 to 5.96%; soluble sugars with an average of 4.22%, varying between 3.61 to 5.03%; an average ether extract of 5.03%, with values ranging from 3.76 to 6.48%; and crude protein with an average of 17.75%, ranging from 22.88% to 14.93% [21]. These results may be considered lower compared to those observed in this study.

Due to their metabolic and physiological origins, the chemical compounds in coffee beans may exhibit correlations with each other. Out of 45 possible phenotypic correlations between ten characteristics, 12 were significant (Table 2). The aqueous extract content showed a positive and significant phenotypic correlation with ash content (r_pe_ = 0.36 **), acidity (r_pe_ = 0.33 **), total soluble solids (r_pe_ = 0.24 *), and phenolic compounds (r_pe_ = 0.42**). Ash content had a positive and significant correlation with acidity (r_pe_ = 0.42), phenolic compounds (r_pe_ = 0.29 *), and reducing sugars (r_pe_ = 0.43 **). Acidity displayed a significant negative correlation with pH (r = −0.39 **) and positive correlations with proteins (r_pe_ = 0.35 **), phenolic compounds (r_pe_ = 0.60 **), and reducing sugars (r_pe_ = 0.36 **). pH showed a significant negative correlation with protein content (r_pe_ = −0.25 *).

Among these phenotypic correlations, approximately 66% were significant in terms of genotypic correlation (Table 2), with similar signs and close magnitudes. This indicates that genetic factors had a stronger influence on the association between traits compared to environmental factors, as phenotypic correlations are derived from measurements affected by both genetic and environmental factors [24].

Unlike the phenotypic correlation estimates, the traits TTA × TSS (r_ge_ = 0.27 ^+^), TCP × TPC (r_ge_ = 0.44 ^++^), and TPC × TRS (r_ge_ = 0.48 ^++^) showed significant correlations solely at the genotypic level (Table 2). Although simple correlations between AE × TSS, TA × TPC, TTS × TPC, and pH × TCP indicated significant associations, the genotypic correlation estimates suggest that these associations are not due to genetic effects.

The significant association between TTA and AE indicates that the solubility of organic acids present in coffee beans is enhanced when they come into contact with boiling water. Similarly, there is a correlation between TPC and AE. Another important association found in this study was between TTA and TCP. Studying the amino acid profile in Robusta coffee, found high levels of glutamic and aspartic acids, confirming the direct proportional relationship between these variables [25].

In the dispersion of the first two principal components, genotypes that are closer together are more similar across all evaluated physicochemical traits simultaneously (Figure 3). The projection of variables onto this dispersion shows that, except for ether extract (EE), genotypes located in the right quadrants generally have higher average values for the traits assessed.

The ideal references for maximum and minimum performance were identified within this dispersion (Figure 3). The genotypes BRS3193, AS1, and AS2 were positioned near the high-performance ideotype, whereas N2, GB7, R22, and WP6 were closer to the low-performance ideal. The BRS3193 cultivar exhibited high levels of TTA, TSS, TPC, SS, and NRS. The AS1 genotype was notable for its high SST content, and the AS2 genotype for its high AE content.

Among the various cultivars, BRS2357 displayed high levels of TCP and TSS, while BRS2299 had a notably high TSS content. Of the publicly available genotypes, GJ30 was distinguished by its high pH and EE, whereas CA1 was noted for its high TTA and low ratio. AS3 was prominent for its elevated levels of TSS, SS, and NRS, while GJ25 showed high contents of TSS, SS, and NRS along with a lower pH. AS7 excelled in both TTA and NRS, and AR106 had high levels of TSS and NRS with a lower pH. GJ8 was recognized for its high TRS and low EA, and GJ3 had the highest TA content. Among the clones from Embrapa’s active germplasm bank in RO, BAG22 and BAG38 stood out for their high TSS levels, while BAG19 was notable for its high SS, NRS, and elevated pH.

A bean with distinct characteristics, such as higher acidity, astringency, and elevated sugar concentration, can be obtained from the cultivar BRS3193, as well as from AS3 and GJ25. These genotypes produce beans with greater sweetness and high levels of water-soluble substances. In contrast, GJ8 has lower solubility in boiling water but features higher levels of reducing sugars, such as glucose and fructose, in its composition.

In addition to being responsible for the sweet flavor of the beverage, sugars are precursors of taste and aroma, reacting in various ways (fragmentation, caramelization, or interaction with amino acids) [26,27]. *C. canephora* tends to achieve higher scores in cup tastings, demonstrating a superior metabolic profile when its soluble constituent concentrations are elevated [5,28]. In this study, we compared the gains from selection across a range of characteristics, focusing on the primary trait of total sugar content.

A genotype suitable for cultivation should display a range of favorable traits. Selection can be based on a single key trait or by using a selection index that simultaneously evaluates multiple traits. In this study, selecting for the primary trait (SS) resulted in a total gain of 32.2% (Table 3). This gain was lower compared to the gains achieved using other selection indices, which showed higher magnitude estimates.

The genotype × ideotype index, which measures the Euclidean distance between the studied genotypes and an ideal plant with maximum performance, yielded the highest estimated gain (SG = 58.86). The Smith & Hazel index, which combines linear traits, showed the second highest gain estimate (SG = 57.19). The Mulamba and Mock index, which aggregates the rankings of genotypes based on their genetic values for each trait, provided the third highest gain estimate (SG = 52.96) (Table 3).

Among the genotypes selected for the primary trait, only genotype BAG19, based on the projection of variables onto the PCA dispersion, is not positioned in the right-hand quadrant. In this quadrant, the cultivar BRS3193 and the clones AS3, AS7, AR106, and LB88 were selected in one or more selection indices. This suggests that these accessions tend to exhibit not only sweetness in their green beans but also a range of other favorable physicochemical characteristics.

In the case of the cultivar BRS3193, this genotype is noted for its commercial beverage quality and less pronounced nuances [4]. It had some desirable chemical characteristics, including a trigonelline content of 0.85%, chlorogenic acid of 5.48%, and caffeine content exceeding 2.70% [5]. AS7 has an average sensory score above 82 points, while AR106 has a score close to 80 points [29].

Among the other ranked genotypes, the cultivar BRS2299 and clone CA1 were selected by all selection indices for their high potential in physicochemical bean quality. They were followed by GJ30, BAG22, AS1, and BAG19, which were selected in two indices, and subsequently by GJ8, BRS3213, AS6, and BRS2357, which were selected in only one index.

Selecting genotypes based on the total sugar content (Table 3) in green coffee beans tends to favor materials with desirable characteristics for the industry, such as high water solubility. However, the clone AS7 is an exception, as it maintained an average value of EA below the mean (Table 4).

In general (Table 4), the 19 genotypes selected using various strategies demonstrate potential for grain production with high levels of TRS and TCP. This promotes the formation of melanoidins and other precursors through the Maillard reaction during roasting. With the exception of BAG19 and LB88, all other genotypes exhibit high levels of TPC, regardless of the selection strategy employed.

The concentration of physicochemical compounds in green coffee beans leads to constituents that are crucial for the quality of the beverage after roasting [30]. And the interactions between them are associated with the quality of the beverage [31]. This includes compounds that are predominantly found in the soluble fraction, such as sugars, proteins, and phenolic compounds. These compounds not only act as precursors to flavor and aroma [32,33] but also contribute to the body of the Beverage characteristic highly valued in the soluble coffee industry.

Table 5 shows the classification of genotypes based on the combined concentration of these three important classes of chemical compounds. According to this classification, which uses Skott-Knott mean clustering, the materials in this study were divided into 5 groups, with concentrations ranging from 23.44% to 33.12%. Higher values are observed for BRS3193 and BRS2357, which are plants with high potential for producing substances that could enhance the value of coffee beans. In contrast, lower values are found in the materials BG180, SK41, WP6, AS10, BAG24, BRS3137, BRS2336, BAG23, N2, and GJ20.

In this study, we assessed the genetic variability of coffee genotypes grown in Western Amazonia, focusing on their physicochemical characteristics across two harvests. The results revealed significant genetic diversity among the materials, identifying a group of promising genotypes for breeding programs and enabling the selection of superior genotypes with desirable traits. Selecting genotypes based on the total sugar content in their green beans tends to favor the development of materials with desirable traits for the industry, such as good solubility in water. Genetic progress estimates indicate a strong potential for successfully selecting plants that exhibit a range of superior quality attributes, rather than focusing on just one trait. This includes high levels of sugars, proteins, lipids, and phenolic compounds in their green beans. Consequently, materials selected for one or more of these traits are highly promising and could be introduced into Rondônia plantations by coffee growers for further genetic improvement.

## 3. Materials and Methods

### 3.1. Field Experiment and Sample Collection

In January 2019, a clonal competition test was installed in the experimental field of the Brazilian Agricultural Research Corporation (Embrapa) at 8°48′05.5″ S and 63°51′02.7″ W at 88 m above sea level. The predominant climate in the region is tropical rainy with dry winter, type “Am” (Köppen), with an average temperature of 26.0 °C and average annual precipitation of 2095 mm. September is the hottest month of the year (27.1 °C) and May is the coldest month (24.9 °C) (Figure 1). In this environment, the chemical characteristics of the soil at a depth of 0 to 20 cm are pH, 5.40; P, 2.00 mg dm^−3^; K, 0.09 cmolc dm^−3^; Ca, 1.48 cmolc dm^−3^; Mg, 1.02 cmolc dm^−3^; Al + H, 13.53 cmolc dm^−3^; Al, 0.87 cmolc dm^−3^; MO, 50.90 g kg^−1^ and V, 16.00%.

In this study, 68 genotypes were assessed (Table 6), including registered cultivars (10) and clones marketed in the public domain, widely cultivated in the Amazon region (58). The evaluations were conducted over two harvests, in the 2020/2021 and 2021/2022 growing seasons, on plants that were 28 and 40 months old, respectively. These cultivars, bearing the ‘BRS’ prefix, are categorized into three distinct compatibility groups and exhibit diverse maturation cycles, including early, intermediate, and late maturation stages [4].

The harvest period was carried out between April and August, considering the maturation cycle of each genotype. To ensure that each genotype was adequately represented, the samples were composed of a mixture of fruits harvested when each plant had at least 70% cherry fruits. The coffee fruits were selected at the cherry stage and washed to remove impurities and defects. The fruit was allowed to dry naturally until the samples reached 11–12% moisture. After drying, the fruits were peeled and the green coffee beans were sieved.

### 3.2. Physicochemical Analysis

The green coffee beans were ground using a blade mill with a 20-mesh sieve. Their characterization was conducted at the Food Science and Technology Laboratory (LCTA) of the Federal Institute of Education, Science, and Technology of Rondônia—Campus Jaru.

For moisture analysis, 4 g of the sample were weighed in a metal capsule and analyzed using a gravimetric technique. This involved heating the sample in a drying oven at 105 °C until a constant weight was reached, measuring the mass loss to determine moisture content [34]. The results were then standardized to a dry basis.

To determine Total Ash (TA), 2 g of the sample were weighed into a porcelain crucible and heated at 550 °C in a muffle furnace [34].

For the analysis aqueous extract (AE), 2 g of the sample were weighed and mixed with 200 mL of hot distilled water, then boiled for 1 h under reflux. The mixture was transferred to a 500 mL volumetric flask. The solid residue in the extraction flask was washed with 100 mL of hot distilled water, and this wash was combined with the remaining extract. After the sample cooled to room temperature, distilled water at 25 °C was added to bring the total volume to 500 mL. The solution was then filtered, and the percentage of aqueous extract was determined gravimetrically [34].

The Total Soluble Solids (TSS) content of the sample was determined by direct measurement using a bench-top ABBE refractometer [34]. For the analysis, 1 g of the sample was mixed with 10 mL of distilled water at room temperature. The samples were centrifuged at 4000 rpm, and then the TSS value was determined and expressed as a percentage.

For pH analysis, 5 g of the sample were weighed and mixed with 50 mL of distilled water. The mixture was stirred for 1 h at 150 rpm using a shaker table. After filtration, the pH was measured with a digital pH meter at room temperature [34]. For this analysis Total Titratable Acidity (TTA), 1 g of the sample was mixed with 100 mL of distilled water and stirred for 1 h at 150 rpm on a shaker table. The mixture was then filtered, and the filtrate was titrated with 0.1 mol·L^−1^ NaOH solution until reaching a pH of 8.1–8.2 at room temperature. The TTA was expressed as the volume of 0.1 mol·L^−1^ NaOH solution used per 100 g of sample [35]. 

To determine Total Crude Protein (TCP), the modified Kjeldahl method was used. For this analysis, 1 g of the dried sample was treated with 7 mL of sulfuric acid and 2.5 g of a catalytic mixture. The results were expressed as a percentage of TCP. The Ether Extract (EE) content was determined using the Soxhlet method. For this, 2 g of the dried sample were placed in a filter paper and contained in a cellulose cartridge, and extracted with petroleum ether. The results were expressed as a percentage of EE [35].

To determine the Soluble Sugars (SS), Total Reducing Sugars (TRS), and Total Phenolic Compounds (TPC), 2.5 g of the samples were weighed, dissolved in 50 mL of distilled water, stirred in a Dubnoff bath for 2 h at 40 °C, and then filtered [36].

The concentration of SS was determined using the Anthrone method, with results expressed as % glucose at a wavelength of 630 nm. The content of TRS was measured using the Somogyi-Nelson method, with values expressed as % glucose at a wavelength of 540 nm [37]. The determination of TPC was carried out using the Folin-Ciocalteu method, and the values were expressed as grams of gallic acid equivalents per 100 g of sample at a wavelength of 760 nm [35]

The Ratio was determined by the ratio of total soluble solids to total titratable acidity. Non-reducing sugars were calculated as the difference between total soluble sugars and total reducing sugars.

### 3.3. Statistical Analyzes

The significance of the clone effects and the homogeneity of residual variances was verified before the combined analysis to quantify the GY interaction effect [38,39].
$$Y_{ijk}=m+G_{i}+Y_{j}+GY_{ij}+e_{ijk}$$
where $Y_{ijk}$ refers to the observation of the ith genotype in the jth measurement, m is the experimental average, $G_{i}$ is the effect of the ith genotype, $Y_{j}$ is the effect of the jth measurement, $GY_{ij}$ is the effect of the interaction between the ith genotype and the jth measurement, and $e_{ijk}$ is the experimental error.

From the estimates of the mean square expected values, repeatability was estimated as follows [14]:$$r=\frac{C\hat{O}V(Y_{ij}Y_{ij\prime })}{\sqrt{\hat{V}\left(Y_{ij}\right)\hat{V}(Y_{ij\prime })}}=\frac{\sigma_{p}^{2}}{\sigma_{p}^{2}+\sigma_{et}^{2}}$$
where: r is the repeatability coefficient; $\sigma_{p}^{2}$ is the genotypic variance combined with the variance of permanent environmental effects; $\sigma_{et}^{2}$ is the temporary environmental variance associated with experimental error. Associations between the first and second measurements were interpreted using Pearson correlation estimates [17].

Genetic progress was quantified, considering direct gains, correlated response, and the use of selection indices. The correlated response, which assesses changes in traits associated with selection for a primary characteristic, was estimated by considering evaluations across both harvests, following the expression [16]:$$\left(y/x\right)=k\cdot r_{\left(x,y\right)}\cdot h_{x}\cdot h_{y}\cdot \sigma_{y}$$

$R\left(y/x\right)$: indirect genetic gain in a trait y as a result of selection for a trait x, k: standardized selection differential, $r_{\left(x,y\right)}$: correlation between traits x and y, $h_{x}$: heritability of trait x, $h_{y}$: heritability of trait y, $\sigma_{y}$: phenotypic standard deviation of trait y.

Genotypic values were employed to quantify genetic progress using the index based on the sum of ranks [40], the Smith & Hazel index [41] and the genotype-ideotype index [38].

The rank sum index, involves summing the ranks of genotypes, which are ordered based on their genetic values for each trait [40]. Genotype classification is then determined by arranging them in descending order of their genetic values for the evaluated traits.

The classical index comprises a linear combination of several economically significant traits, with weighting coefficients estimated to maximize the correlation between the index and the genotypic aggregate [41]. This aggregate is determined by another linear combination of genetic values weighted by their respective economic values [38]. The expected gain for trait y, when selection is performed based on the index, is given by the expression:$${∆g}_{y(x)}={DS}_{y(x)}h_{y}^{2}$$

${∆g}_{y(x)}$: expected gain for trait y when selection is practiced using the index, ${DS}_{y(x)}$: selection differential of trait y compared to index x, $h_{y}^{2}$: heritability of trait y.

In the genotype-ideotype index [38], estimated distances between genotypes and reference values are considered, defined by the observed maximums and minimums, as per the expression:$$G_{i}={\left[1/n\sum_{j=1}^{n}{\left(x_{ij}-m_{j}\right)}^{2}\right]}^{0.1}$$
where: G_i_ is the genotype-ideotype distance; x_ij_ is the score of the principal component analysis technique for the i-th genotype on the j-th principal component; and m_j_ is the score associated with the ideal reference on the j-th principal component.

Genetic diversity among the genotypes was assessed by analyzing the dispersion of the first two principal components, which were derived from the genotypic values of all evaluated physicochemical characteristics. All statistical analyses were performed using the software GENES, version 1990.2023.93 [34] and Selegen, version 2020 [16].

## 4. Conclusions

The high genetic diversity of green coffee beans cultivated in the Western Amazon favors the selection of plants with physicochemical characteristics of interest both to the soluble coffee industry and for beverage quality. Selection using selection index favored superior selection gains compared to direct selection, indicating that genotypes such as BRS3193 and CA1, selected by various indices, tend to present more than one favorable characteristic for quality attributes in their green beans. The findings of this study are likely to favor the genetic improvement of the *C. canephora* species, aiming to produce new materials that can meet both the industrial niche and beverage quality standards.

## Figures and Tables

**Figure 1 plants-13-02780-f001:**
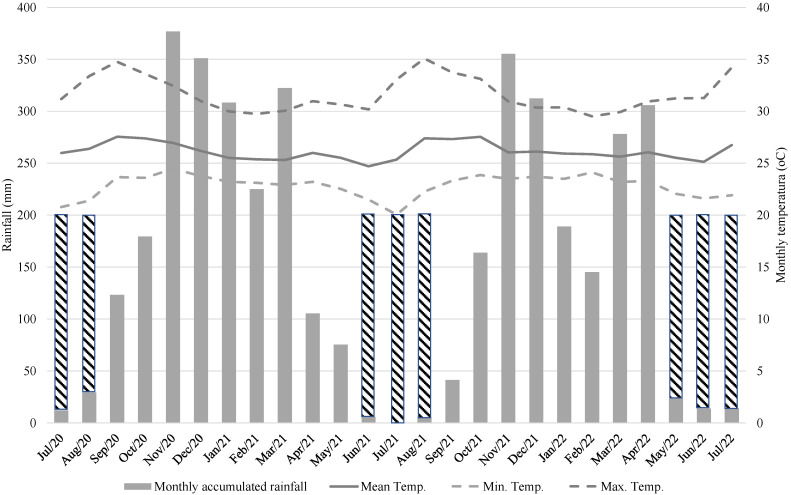
Monthly accumulated rainfall, maximum, mean and minimum temperatures from July 2020 to July 2022 in the environment of Porto Velho-RO, recorded in Ambient Weather WS2902. The razored area represents supplementary irrigation managed in the months of June, July, and August.

**Figure 2 plants-13-02780-f002:**
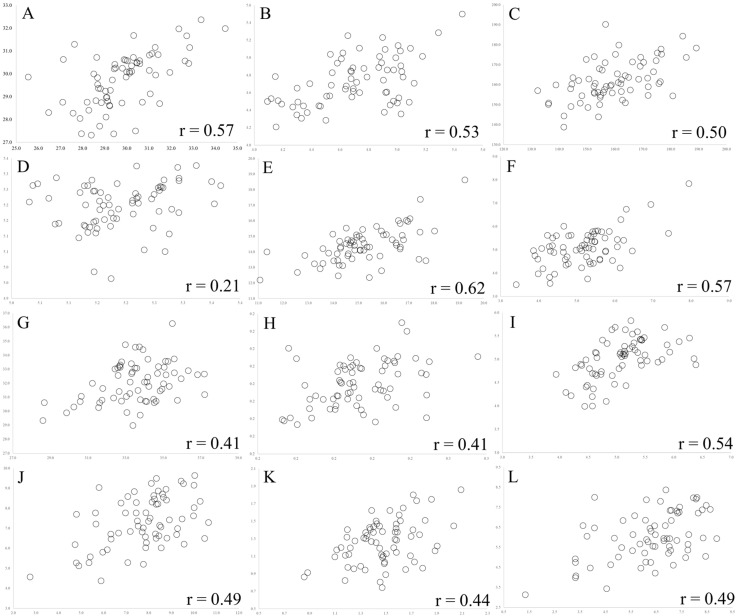
Analysis of repeated measures between the physicochemical characteristics of the most cultivated genotypes in the Western Amazon, Brazil, evaluated in two measurements, the 2020–2021 and 2021–2022 harvests. (**A**): aqueous extract (%d.b.), (**B**): total ash (%d.b.), (**C**): total titratable acidity (mL NaOH 0.1 mol·L·100 g^−1^ d.b.), (**D**): hydrogen potential (d.b.), (**E**): total crude protein (%d.b.), (**F**): ether extract (%d.b.), (**G**): total soluble solids (%d.b.), (**H**): ratio (% total soluble solids/mL of NaOH d.b.), (**I**): total phenolic compounds (g gallic acid eq./100 g of the sample d.b.), (**J**): soluble sugars (%d.b.), (**K**): total reducing sugars (%d.b.), (**L**): non-reducing sugars (%d.b.).

**Figure 3 plants-13-02780-f003:**
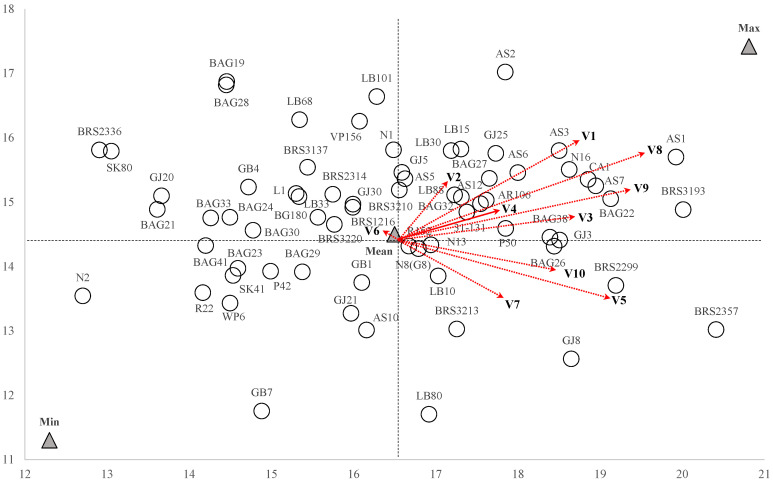
Graphical dispersion of the physicochemical characteristics of the most cultivated genotypes in the Western Amazon, Brazil, evaluated in two measurements, the 2020–2021 and 2021–2022 harvests, with the reference axes for CP. The red vectors represent the projection of the characteristics: V1: aqueous extract; V2: total ash; V3: total titratable acidity; V4: pH; V5: total crude protein; V6: ethereal extract; V7: total soluble solids; V8: total phenolic compounds; V9: soluble sugars; V10: total reducing sugars.

**Table 1 plants-13-02780-t001:** Summary of variance analyzes of the physicochemical characteristics of the most cultivated genotypes in the Western Amazon, Brazil, evaluated in two measurements, the 2020–2021 and 2021–2022 harvests.

**SV**	**DF**	**AE**	**TA**	**TTA**	**pH**	**TCP**	**EE**
Genotypes (G)	67	3.38 **	3.23 **	2.80 **	1.53 *	3.90 **	3.62 **
Years (Y)	1	5.15 *	18.75 **	54.48 **	14.96 **	546.85 **	12.73 **
GxE	67	36.45 **	36.79 **	11.83 **	36.12 **	18.96 **	11.92 **
Residue	272						
Sum	407						
Mean 1° year		29.81 a	4.70 a	158.77 b	5.19 a	15.20 a	5.18 a
Mean 2° year		29.75 a	4.73 a	161.91 a	5.19 a	14.43 b	5.08 a
Mean		29.78	4.72	160.34	5.19	14.81	5.13
CVe		0.93	1.22	2.68	0.37	2.25	5.15
r		70.48	69.1	64.34	34.91	74.4	72.41
CVg		3.57	4.53	5.06	0.67	6.86	11.78
CVg/Cve		3.84	3.71	1.89	1.81	3.05	2.29
**SV**	**DF**	**TSS**	**Ratio**	**TPC**	**SS**	**TRS**	**NRS**
Genotypes (G)	67	2.34 **	2.26 **	3.18 **	2.84 **	2.58 **	2.78 **
Years (Y)	1	183.36 **	186.31 **	104.44 **	120.06 **	1097.38 **	38.53 **
GxE	67	3.75 **	6.02 **	63.29 **	19.35 **	24.45 **	18.18 **
Residue	272						
Sum	407						
Mean 1° year		33.53 a	0.21 a	5.07 a	7.75 a	1.47 a	6.29 a
Mean 2° year		31.97 b	0.20 b	4.99 a	7.31 b	1.26 b	6.04 a
Mean		32.75	0.20	5.03	7.53	1.36	6.16
CVe		3.57	4.69	1.45	5.47	4.71	6.71
r		57.33	55.93	68.63	64.79	61.34	64.03
CVg		3.27	5.3	6.98	13.34	11.99	15.6
CVg/Cve		0.92	1.13	4.81	2.44	2.55	2.32

* significant at 5% probability. ** significant at 1% probability. SV: source of variation, DF: degrees of freedom, CVe: experimental coefficient of variation, r: repeatability coefficient, CVg: genetic coefficient of variation. d.b.: dry base. AE: aqueous extract (%d.b.), TA: total ash (%d.b.), TTA: total titratable acidity (mL NaOH 0.1 mol·L·100 g^−1^ d.b.), pH: hydrogen potential (d.b.), TCP: total crude protein (%d.b.), EE: ether extract (%d.b.), TSS: total soluble solids (%d.b.), ratio (% total soluble solids/mL of NaOH d.b.), TPC: total phenolic compounds (g gallic acid eq./100 g of the sample d.b.), SS: soluble sugars (%d.b.), TRS: total reducing sugars (%d.b.), NRS: non-reducing sugars (%d.b.). Means followed by equal letter do not differ, by Scott-Knott test, at 5% probability.

**Table 2 plants-13-02780-t002:** Estimates of phenotypic and genotypic correlation coefficients referring to the physicochemical characteristics of the most cultivated genotypes in the Western Amazon, Brazil, evaluated in two measurements, the 2020–2021 and 2021–2022 harvests.

n	Variable	rpe	rge	n	Variable	rpe	rge
1	AE × TA	0.36 **	0.37 ^++^	24	TTA × TRS	0.36 **	0.40 ^++^
2	AE × TTA	0.33 **	0.34 ^++^	25	pH × TCP	−0.25 *	−0.30 ^NS^
3	AE × pH	−0.16 ^NS^	−0.17 ^NS^	26	pH × EE	0.10 ^NS^	0.10 ^NS^
4	AE × TCP	0.14 ^NS^	0.13 ^NS^	27	pH × TSS	−0.25 ^NS^	−0.30 ^NS^
5	AE × EE	−0.18 ^NS^	−0.18 ^NS^	28	pH × TPC	−0.25 ^NS^	−0.31 ^NS^
6	AE × TSS	0.24 *	0.36 ^NS^	29	pH × SS	−0.08 ^NS^	−0.06 ^NS^
7	AE × TPC	0.42 **	0.44 ^++^	30	pH × TRS	−0.19 ^NS^	−0.22 ^NS^
8	AE × SS	0.15 ^NS^	0.14 ^NS^	31	TCP × EE	−0.09 ^NS^	−0.09 ^NS^
9	AE × TRS	0.17 ^NS^	0.16 ^NS^	32	TCP × TSS	0.50 ^NS^	0.52 ^NS^
10	TA × TTA	0.42 **	0.44 ^++^	33	TCP × TPC	0.42 ^NS^	0.44 ^++^
11	TA × pH	−0.04 ^NS^	−0.05 ^NS^	34	TCP × SS	−0.11 ^NS^	−0.13 ^NS^
12	TA × TCP	0.22 ^NS^	0.22 ^NS^	35	TCP × TRS	0.49 ^NS^	0.52 ^NS^
13	TA × EE	−0.09 ^NS^	−0.11 ^NS^	36	EE × SST	−0.04 ^NS^	−0.03 ^NS^
14	TA × TSS	0.20 ^NS^	0.21 ^NS^	37	EE × TPC	−0.04 ^NS^	−0.04 ^NS^
15	TA × TPC	0.29 *	0.37 ^NS^	38	EE × SS	0.01 ^NS^	0.01 ^NS^
16	TA × SS	0.06 ^NS^	0.06 ^NS^	39	EE × TRS	−0.02 ^NS^	−0.03 ^NS^
17	TA × TRS	0.43 **	0.46 ^++^	40	TSS × TPC	0.25 ^NS^	0.35 ^NS^
18	TTA × pH	−0.39 **	−0.41 ^++^	41	TSS × SS	0.03 ^NS^	0.03 ^NS^
19	TTA × TCP	0.35 **	0.38 ^++^	42	TSS × TRS	0.37 ^NS^	0.46 ^NS^
20	TTA × EE	0.05 ^NS^	0.07 ^NS^	43	TPC × SS	0.09 ^NS^	0.09 ^NS^
21	TTA × TSS	0.23 ^NS^	0.27 ^+^	44	TPC × TRS	0.46 ^NS^	0.48 ^++^
22	TTA × TPC	0.60 **	0.68 ^NS^	45	SS × TRS	0.21 ^NS^	0.22 ^NS^
23	TTA × SS	0.16 ^NS^	0.14 ^NS^				

* fhenotypic correlation 5% of probability. ** fhenotypic correlation 1% of probability. ^NS^ not significant. ^+^ genotypic correlation 5% of probability. ^++^ genotypic correlation 1% of probability. rpe: phenotypic coefficient correlation. rge: genotypic coefficient correlation. d.b.: dry base. AE: aqueous extract (%d.b.), TA: total ash (%d.b.), TTA: total titratable acidity (mL NaOH 0.1 mol·L·100 g^−1^ d.b.), pH: hydrogen potential (d.b.), TCP: total crude protein (%b.s.), EE: ether extract (%d.b.), TSS: total soluble solids (%d.b.), TPC: total phenolic compounds (g gallic acid eq./100 g of the sample d.b.), SS: soluble sugars (%d.b.), TRS: total reducing sugars (%d.b.).

**Table 3 plants-13-02780-t003:** Estimates of genetic progress (%) obtained by selection indexes and by univariate direct and indirect selection for the physicochemical characteristics of the most cultivated genotypes in the Western Amazon, Brazil, evaluated in two measurements, the 2020–2021 and 2021–2022 harvests.

	Estimates of Progress with Selection											
Index		AE	TA	TTA	pH	TCP	EE	TSS	TPC	SS	TRS	SG
Direct selection ^#1^	0.89		0.52	1.3	−0.1	−1.54	0.28	0.18	1.13	24.66	4.88	32.2
Genotype Ideotype ^#2^	−1.43		2.44	7.01	−0.19	7.11	14.06	2.5	5.85	6.58	14.93	58.86
Smith & Razel ^#3^	2.29		1.63	9.23	−0.38	4.94	6.51	2.43	8.86	10.84	10.84	57.19
Mulamba & Mock ^#4^	3.14		3.02	7.21	−0.94	7.92	1.78	3.89	8.63	9.47	8.84	52.96
	Ordering of genotypes selected by each index											
Genotypes	#1			#2			#3			#4		
BRS3193	1			1			2			1		
AS3	2			NS			4			3		
GJ25	3			NS			NS			NS		
LB30	4			NS			NS			NS		
BAG19	5			NS			NS			NS		
AS7	6			7			3			NS		
AS2	7			NS			NS			NS		
AR106	8			NS			NS			6		
LB88	9			5			NS			NS		
CA1	NS			2			1			7		
GJ30	NS			3			5			NS		
BRS2299	NS			4			6			5		
GJ8	NS			6			NS			NS		
BRS3213	NS			8			NS			NS		
BAG38	NS			9			NS			9		
AS6	NS			NS			7			NS		
BAG22	NS			NS			8			4		
AS1	NS			NS			9			2		
BRS2357	NS			NS			NS			8		

d.b.: dry base. AE: aqueous extract (%d.b.), TA: total ash (%d.b.), TTA: total titratable acidity (mL NaOH 0.1 mol·L·100 g^−1^ d.b.), pH: hydrogen potential (d.b.), TCP: total crude protein (%d.b.), EE: ether extract (%d.b.), TSS: total soluble solids (%d.b.), TPC: total phenolic compounds (g gallic acid eq./100 g of the sample d.b.), SS: soluble sugars (% d.b.), TRS: total reducing sugars (%d.b.), SG: gain from selection, NS: not select.

**Table 4 plants-13-02780-t004:** Average performance across two harvests for the 19 genotypes selected using different strategies. Evaluations of all genotypes can be found in the Appendix A section.

Genotypes	AE	TA	TTA	pH	TCP	EE	TSS	Ratio	TPC	SS	TRS	NRS
BRS3193	30.24 c	4.78 c	179.60 a	5.11 d	17.42 b	5.40 e	33.96 a	0.19 b	5.86 a	9.84 a	1.47 b	8.03 a
AS3	31.21 b	4.74 c	175.57 a	5.17 c	15.55 c	5.23 e	33.87 a	0.19 b	5.33 a	9.57 a	1.39 b	7.84 a
GJ25	30.61 c	4.54 d	170.59 b	5.06 d	13.89 d	4.72 f	33.94 a	0.20 a	5.10 b	9.39 a	1.42 b	7.97 a
LB30	30.47 c	4.82 c	154.16 d	5.14 c	14.93 c	5.04 e	32.72 b	0.21 a	5.41 a	9.38 a	1.31 b	7.99 a
BAG19	31.00 b	4.67 c	148.94 d	5.31 a	13.74 d	4.62 f	30.91 c	0.20 a	4.45 c	9.31 a	1.30 b	8.01 a
AS7	28.96 d	4.87 b	183.80 a	5.16 c	15.56 c	4.87 f	33.00 b	0.19 a	5.25 a	9.02 a	1.80 a	7.68 a
AS2	33.21 a	4.95 b	161.34 c	5.23 b	14.14 d	4.63 f	32.92 b	0.20 a	5.11 b	8.96 a	1.55 b	7.41 a
AR106	31.12 b	4.65 c	163.74 c	5.06 d	14.87 c	6.51 c	34.34 a	0.21 a	5.10 b	8.91 a	1.29 b	7.62 a
LB88	30.11 c	4.99 b	161.24 c	5.26 b	15.68 c	6.22 c	33.34 b	0.20 a	4.64 c	8.85 a	1.73 a	7.27 a
CA1	30.07 c	5.28 a	184.33 a	5.17 c	15.02 c	6.94 b	33.28 b	0.17 b	5.47 a	7.43 b	1.45 b	6.01 b
GJ30	29.01 d	4.45 d	176.01 a	5.32 a	14.88 c	7.89 a	32.57 b	0.18 b	5.27 a	8.31 a	1.38 b	6.96 a
BRS2299	30.03 c	4.76 c	173.40 b	5.13 c	16.30 b	5.08 e	35.87 a	0.20 a	5.53 a	7.78 b	1.48 b	6.29 b
GJ8	27.70 e	4.91 b	168.83 b	5.19 c	16.36 b	5.75 d	33.33 b	0.19 a	5.63 a	5.49 c	1.97 a	4.03 c
BRS3213	28.91 d	4.45 d	154.32 d	5.12 c	16.68 b	5.70 d	34.15 a	0.22 a	5.08 b	8.13 b	1.57 b	6.55 b
BAG38	28.87 d	5.06 b	173.41 b	5.16 c	15.53 c	5.35 e	34.16 a	0.20 a	5.39 a	7.82 b	1.46 b	6.36 b
AS6	30.58 c	4.76 c	177.20 a	5.18 c	14.66 c	4.92 f	32.65 b	0.18 b	5.47 a	7.51 b	1.49 b	6.01 b
BAG22	31.59 b	4.76 c	171.49 b	5.15 c	15.53 c	4.77 f	34.09 a	0.20 a	5.68 a	7.88 b	1.56 b	6.32 b
AS1	32.86 a	4.79 c	168.90 b	5.13 c	15.42 c	4.32 f	34.13 a	0.20 a	5.73 a	8.48 a	1.74 a	6.74 b
BRS2357	31.01 b	5.00 b	167.67 b	5.14 c	18.93 a	3.45 g	34.91 a	0.21 a	5.39 a	7.08 b	1.66 a	5.42 b

d.b.: dry base. AE: aqueous extract (%d.b.), TA: total ash (%d.b.), TTA: total titratable acidity (mL NaOH 0.1 mol·L·100 g^−1^ d.b.), pH: hydrogen potential (d.b.), TCP: total crude protein (%d.b.), EE: ether extract (%d.b.), TSS: total soluble solids (%d.b.), ratio (% total soluble solids/mL of NaOH d.b.), TPC: total phenolic compounds (g gallic acid eq./100 g of the sample d.b.), SS: soluble sugars (%d.b.), TRS: total reducing sugars (%d.b.), NRS: non-reducing sugars (%d.b.). Means followed by equal letter do not differ, by Scott-Knott test, at 5% probability.

**Table 5 plants-13-02780-t005:** Combined concentration of total crude protein, total phenolic compounds, and total soluble sugars by Scott-Knott test, at 5% probability, of the most cultivated genotypes in the Western Amazon, Brazil, evaluated in two measurements, the 2020–2021 and 2021–2022 harvests.

Concentration (%d.b.)	Genotypes
>31.4	BRS3193; BRS2357
>29.6	AS3; BRS3213; P50; AS7; LB30; AS1; BRS2299
>27.4	31–131; LB88; BAG22; AR106; BAG38; BAG26; LB101; N13; GJ30; BAG41; GJ25; AS2; LB15; AS5; LB68; GJ5; N8(G8); CA1; GJ21; N16; L1; BAG29; AS6; BAG30; BAG19; GJ8; BAG27
>25.6	BRS3210; GB1; AS12; LB80; R152; SK80; BAG28; P42; VP156; BRS2314; BRS1216; BAG32; LB10; BAG33; BAG21; GB7; BRS3220; GB4; GJ3; LB33; N1; R22
>23.4	BG180; SK41; WP6; AS10; BAG24; BRS3137; BRS2336; BAG23; N2; GJ20

d.b.: dry base.

**Table 6 plants-13-02780-t006:** Coffee genotypes evaluated for the physicochemical characteristics of green beans from from the active germplasm bank (BAG), cultivars developed by Embrapa (BRS) and materials in the public domain grown in Western Amazon, Brazil.

No.	Genotype	Origin	No.	Genotype	Origin	No.	Genotype	Origin
1	BAG19	Embrapa ^1^	24	BRS3220	Embrapa ^1^	47	GB7	Gilberto Boon ^2^
2	BAG21	Embrapa ^1^	25	AS1	Ademar Schmidt ^2^	48	LB10	Laerte Braun ^5^
3	BAG22	Embrapa ^1^	26	AS2	Ademar Schmidt ^2^	49	LB15	Laerte Braun ^5^
4	BAG23	Embrapa ^1^	27	AS3	Ademar Schmidt ^2^	50	LB30	Laerte Braun ^5^
5	BAG24	Embrapa ^1^	28	AS5	Ademar Schmidt ^2^	51	LB33	Laerte Braun ^5^
6	BAG26	Embrapa ^1^	29	AS6	Ademar Schmidt ^2^	52	LB68	Laerte Braun ^5^
7	BAG27	Embrapa ^1^	30	AS7	Ademar Schmidt ^2^	53	LB80	Laerte Braun ^5^
8	BAG28	Embrapa ^1^	31	AS10	Ademar Schmidt ^2^	54	LB88	Laerte Braun ^5^
9	BAG29	Embrapa ^1^	32	AS12	Ademar Schmidt ^2^	55	LB110	Laerte Braun ^5^
10	BAG30	Embrapa ^1^	33	L1	Alcides Rosa ^3^	56	N1	Nivaldo Ferreira ^6^
11	BAG32	Embrapa ^1^	34	BG180	Adilson Berger ^3^	57	N2	Nivaldo Ferreira ^6^
12	BAG33	Embrapa ^1^	35	AR106	Aldinei Raasch ^8^	58	N8(G8)	Nivaldo Ferreira ^6^
13	BAG38	Embrapa ^1^	36	CA1	Carlos Alves Silva ^4^	59	N13	Nivaldo Ferreira ^6^
14	BAG41	Embrapa ^1^	37	GJ3	Geraldo Jacomini ^5^	60	N16	Nivaldo Ferreira ^6^
15	BRS1216	Embrapa ^1^	38	GJ5	Geraldo Jacomini ^5^	61	R22	Ronaldo Vitoriano ^2^
16	BRS2299	Embrapa ^1^	39	GJ8	Geraldo Jacomini ^5^	62	R152	Ronaldo G Oliveira ^2^
17	BRS2314	Embrapa ^1^	40	GJ20	Geraldo Jacomini ^5^	63	SK41	Sergio Kalk ^6^
18	BRS2336	Embrapa ^1^	41	GJ21	Geraldo Jacomini ^5^	64	SK80	Sergio Kalk ^6^
19	BRS2357	Embrapa ^1^	42	GJ25	Geraldo Jacomini ^5^	65	VP156	Valdecir Piske ^2^
20	BRS3137	Embrapa ^1^	43	GJ30	Geraldo Jacomini ^5^	66	P50	Valdecir Piske ^2^
21	BRS3193	Embrapa ^1^	44	31–131	Geraldo Jacomini ^5^	67	WP6	Wanderley Peter ^6^
22	BRS3210	Embrapa ^1^	45	GB1	Gilberto Boon ^2^	68	P42	Wanderly Bernabé ^7^
23	BRS3213	Embrapa ^1^	46	GB4	Gilberto Boon ^2^			

^1^ Ouro Preto do Oeste—RO, ^2^ Alta Floresta do Oeste—RO, ^3^ Rolim de Moura—RO, ^4^ Novo Horizonte do Oeste—RO, ^5^ Nova Brasilândia do Oeste—RO, ^6^ Cacoal—RO, ^7^ Alto Alegre dos Parecis, RO, ^8^ São Miguel do Guaporé—RO.

## Data Availability

The original contributions presented in the study are included in the article, further inquiries can be directed to the corresponding author.

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
