# Peer review of "Genetic Variability in the Physicochemical Characteristics of Cultivated Coffea canephora Genotypes"

_plants, 2024, doi:10.3390/plants13192780_

Round 1

Reviewer 1 Report

Comments and Suggestions for Authors

Review of the manuscript titled: Exploring the genetic variability of the physicochemical characteristics of the most cultivated Coffea canephora genotypes.

The study is devoted to an important subject of coffee quality. It uses relevant material and methodology, the results are significant, the study contributes to our knowledge and deserves publication. However, the manuscript is poorly prepared with confusion of the tables and figures. The presentation of the results and their interpretation also deserves improvement.  Perhaps, the results and discussion section can be divided into clearly defined sub-sections.

1.      Field experiment and data collection. The authors refer to two evaluation periods (based on the Table 2 title) 2019-21 and 2021-22. Not clear why the authors call them “evaluation periods” instead of years. If there was only one evaluation in each evaluation period, then the authors can refer to them as years or seasons. If there was more than one evaluation within evaluation periods – then the authors need to explain why they did not use every evaluation as a separate factor. Also, why one evaluation is three years and another two years. This needs to be clearly presented in the methodology. In addition, the authors may like to mention the age of the plants and the harvesting period.

2.      The weather graph does not show 2019 at all – why is it mentioned in the table. May be harvesting time can be added to the weather graph.

3.      Results start with the statement: “Genotype × evaluation (G×E) interactions were significant…”. It would be very useful first to present the mean values for the traits in two seasons and discuss the differences between the seasons from biological, agronomic and meteo perspective prior to presenting the statistical outcomes. The relevant values from Table 2 (Mean 1, Mean 2, Mean, which are not even explained, and related CVs) can be extracted to make new table or convert them to graphs. Once the agronomic performance of the coffee in two seasons is explained – Anova can follow.

4.      Line 62 refers to table with G x E results but it is a table presenting material. Line 260 referes to Table 4 but in fact it is the information from Table 1.

5.      Lines 63-64. “The genotype x evaluation interaction was classified as simple (Figure 1)”. However, Figure 1 presents weather conditions but there is not figure explaining G x E.

6.      Line 79 refers to Table 1 but the information is in Table 2.

7.      Line 76: repeatability is not well explained and, in the table, referred to as “r” – most likely correlation between two evaluations. Is it phenotypic or genotypic correlation? Why some terminology on the text and other in the table.

8.      Lines 80-81. “TCP > AE ≥ EE >  TPC > TA > TTA > SS ≥ NRS > TRS > TSS > ratio > pH”. These abbreviations have not been explained in the text up to now. They are explained at a footnote of Table 2 which is two pages down. Are all these abbreviations needed or some can be described by words?

9.      Lines 94-97: while describing variation for traits we normally start with lower values and then mention the higher values. The authors do it differently. Any reason for this.

10.  The authors tend to explain simple things by long sentences: “The fixed mineral residue, determined by weighing the residue after complete combustion of organic compounds…” – this is simply ash. “…repeated measures between the physicochemical characteristics…” – this is simply correlation. This sophisticated explanation of simple things makes paper difficult to read and follow.

11.  Lines 154-155: “Among these phenotypic correlations, approximately 66% were significant in terms of genotypic correlation (Table 2), with similar signs and close magnitudes”. The Table 2 does not have this data.

12.  There are some words and phrases using non-English: “the coffee beans were sieved, dando origem ao caf verde”, “de probabilidade”.

13.  Line 343: 𝑌𝑗 in the formula and Ej in explanation.

14.  The discussion and conclusions need better focus on practical application of the findings in coffee breeding.

15.  Figure 2 – the traits can be written in the graphs. Is there a need to present all correlations.

16.  The authors state: “The knowledge of the genetic variability, as studied in this work, challenges the long standing paradigm that the C. canephora is associated with lower quality and market value.” In the paper there is nothing on this subject. Perhaps the authors can compare the mean values for the studied traits of the groups of material with different origin.

17.  Lines 262-263: “ These cultivars, bearing the 'BRS' prefix, are categorized into three distinct compatibility groups and exhibit diverse maturation cycles, including early, intermediate, and late maturation stages”. There is no information on the maturity group in Table 1. The superior genotypes identified in the study belong to which group?

18.  The paper deserves the table with the actual data of 68 genotypes performance for the studied traits in the supplement or in the paper body for the superior quality genotypes.

19.  The paper title does not reflect its essence. The authors did not study “genetic”  diversity and there is no information on the genes. This is phenotypic diversity for physicochemical traits using some genetic parameters. “…the most cultivated … genotypes…”. Is it likely that all 68 genotypes are “most cultivated”. Perhaps commonly grown.

Comments on the Quality of English Language

In the review. 

Author Response

Letter to the Editor

We are resubmitting the manuscript entitled “Exploring the Genetic Variability of the Physicochemical Characteristics of the Most Cultivated Coffea canephora Genotypes” for further review following a careful revision.

The authors recognize the challenges involved in producing high-quality scientific work suitable for publication in a journal renowned for its scientific rigor. We have meticulously addressed each of the comments and suggestions received.

We would like to express our gratitude for the detailed feedback provided. All comments have been thoroughly considered in the revised version of the manuscript. At the end of the manuscript, we have addressed each contribution individually.

Revised sections of the text have been highlighted in blue to facilitate the identification of changes.

Thank you for your consideration.

Sincerely,

The authors.

Review:

Reviewer: The study is devoted to an important subject of coffee quality. It uses relevant material and methodology, the results are significant, the study contributes to our knowledge and deserves publication. However, the manuscript is poorly prepared with confusion of the tables and figures. The presentation of the results and their interpretation also deserves improvement.  Perhaps, the results and discussion section can be divided into clearly defined sub-sections.

Authors: We acknowledge the necessary care involved in the development of a scientific work, and we appreciate the comments provided. The work has been carefully revised, including the addition of new tables and a review of the text. To facilitate the identification of the revisions made, we have highlighted the text in blue. The received comments are answered in detail below, along with the indication of the revisions made to the manuscript.

Reviewer 1: Field experiment and data collection. The authors refer to two evaluation periods (based on the Table 2 title) 2019-21 and 2021-22. Not clear why the authors call them “evaluation periods” instead of years. If there was only one evaluation in each evaluation period, then the authors can refer to them as years or seasons. If there was more than one evaluation within evaluation periods – then the authors need to explain why they did not use every evaluation as a separate factor. Also, why one evaluation is three years and another two years. This needs to be clearly presented in the methodology. In addition, the authors may like to mention the age of the plants and the harvesting period.

Authors: We understand that the description of the methodology can be improved. The evaluations were conducted during two harvests, in the agricultural years 2020/2021 and 2021/2022, on plants that were 28 and 40 months old, respectively. All table and figure captions have been revised to better address these issues: Table 1 on Page 3; Figure 2 on Page 4; Table 2 on Page 6; Figure 3 on Page 7; Table 3 on Page 8; Table 5 on Page 10. The text has been revised to better address these issues: Page 3, Lines 100 and 103; Page 10, Line 342; Page 11, Lines 361, 362, 365, and 366.

Reviewer 2: The weather graph does not show 2019 at all – why is it mentioned in the table. May be harvesting time can be added to the weather graph.

Authors: The graph displays the correct information; however, a typographical error was found in the text. The text has been revised to accurately reflect the evaluated periods. All table and figure captions have been updated to address these issues more clearly: Table 1 on Page 3; Figure 2 on Page 4; Table 2 on Page 6; Figure 3 on Page 7; Table 3 on Page 8; Table 5 on Page 10.

Reviewer 3: Results start with the statement: “Genotype × evaluation (G×E) interactions were significant…”. It would be very useful first to present the mean values for the traits in two seasons and discuss the differences between the seasons from biological, agronomic and meteo perspective prior to presenting the statistical outcomes. The relevant values from Table 2 (Mean 1, Mean 2, Mean, which are not even explained, and related CVs) can be extracted to make new table or convert them to graphs. Once the agronomic performance of the coffee in two seasons is explained – Anova can follow.

Authors: We appreciate your observation and agree that presenting the means for each measurement allows for a better consideration of the differences between evaluation periods. This has been addressed on Page 2, Lines 64 to 68 and 75 to 78; and on Page 3, Lines 79 to 87. As noted, we also consider it appropriate to include these estimates, along with the Scott-Knott classification test, in Table 1, Page 3, Line 98.

Reviewer 4:Line 62 refers to table with G x E results but it is a table presenting material. Line 260 referes to Table 4 but in fact it is the information from Table 1.

Authors: We have noted the inappropriate table references in the text and other formatting issues with the tables pointed out in the revised file. All table and figure references have been reviewed. Table 1 on Page 3; Figure 2 on Page 4; Table 2 on Page 6; Figure 3 on Page 7; Table 3 on Page 8; Table 5 on Page 10.

Reviewer 5: Lines 63-64. “The genotype x evaluation interaction was classified as simple (Figure 1)”. However, Figure 1 presents weather conditions but there is not figure explaining G x E.

Line 79 refers to Table 1 but the information is in Table 2.

Authors: All table and figure references have been reviewed. Table 1 on Page 3; Figure 2 on Page 4; Table 2 on Page 6; Figure 3 on Page 7; Table 3 on Page 8; Table 5 on Page 10.

Reviewer 6: Line 79 refers to Table 1, but the information is in Table 2.

 Authors: All table and figure references have been reviewed. Table 1 on Page 3; Figure 2 on Page 4; Table 2 on Page 6; Figure 3 on Page 7; Table 3 on Page 8; Table 5 on Page 10.

Reviewer 7:Line 76: repeatability is not well explained and, in the table, referred to as “r” – most likely correlation between two evaluations. Is it phenotypic or genotypic correlation? Why some terminology on the text and other in the table.

Authors: We also understand that this result can be better presented, emphasizing that the interpretation of this estimate considers the genotypic variance, the variance of permanent environmental effects, and the variance of experimental error. Page 4, Lines 107 to 110.

Reviewer 8: Lines 80-81. “TCP > AE ≥ EE >  TPC > TA > TTA > SS ≥ NRS > TRS > TSS > ratio > pH”. These abbreviations have not been explained in the text up to now. They are explained at a footnote of Table 2 which is two pages down. Are all these abbreviations needed or some can be described by words?

Authors: We also agree that the text can be improved. The names and their respective abbreviations have been introduced at the beginning of the Results and Discussion sections. Page 3, Lines 80 to 84.

Reviewer 9: Lines 94-97: while describing variation for traits we normally start with lower values and then mention the higher values. The authors do it differently. Any reason for this.

 Authors: The presentation of the characteristics has been standardized, starting with the lowest values. Page 5, Lines 139 to 141, 148 and 149, 156 to 159, 160, 162 and 163, 166, 171 to 174.

Reviewer 10: The authors tend to explain simple things by long sentences: “The fixed mineral residue, determined by weighing the residue after complete combustion of organic compounds…” – this is simply ash. “…repeated measures between the physicochemical characteristics…” – this is simply correlation. This sophisticated explanation of simple things makes paper difficult to read and follow.

Authors: We recognize that the text can be improved. The following paragraph has been revised to simplify the interpretation of the results: Page 5, Line 166.

Reviewer 11: Lines 154-155: “Among these phenotypic correlations, approximately 66% were significant in terms of genotypic correlation (Table 2), with similar signs and close magnitudes”. The Table 2 does not have this data.

Authors: All tables and figures have been reviewed, and their numbering has been adjusted according to their citation in the manuscript text. Table 1 on Page 3; Figure 2 on Page 4; Table 2 on Page 6; Figure 3 on Page 7; Table 3 on Page 8; Table 5 on Page 10.

Reviewer 12: There are some words and phrases using non-English: “the coffee beans were sieved, dando origem ao caf� verde”, “de probabilidade”.

Authors: All sentences and words have been properly corrected. Page 11, Line 371; Page 6, Line 190.

Reviewer 13: Line 343: ?? in the formula and Ej in explanation.

 Authors: The text has been revised. Page 12, Line 424.

Reviewer 14: The discussion and conclusions need better focus on practical application of the findings in coffee breeding.

Authors: Once again, we would like to express our gratitude. During the development of our work, we also sought the best way to classify our genotypes according to their potential use. In response to the comments received, we have revised the text and included Table 5, which classifies the genotypes based on their combined concentration of protein, phenolic compounds, and total sugars. Page 10, Lines 310 to 315, 321 to 327.

Reviewer 16: The authors state: “The knowledge of the genetic variability, as studied in this work, challenges the long standing paradigm that the C. canephora is associated with lower quality and market value.” In the paper there is nothing on this subject. Perhaps the authors can compare the mean values for the studied traits of the groups of material with different origin.

Authors: We also agree that the discussion of the results can be improved. We have revised the comparisons between our evaluations and the results observed by different authors. Page 5, Lines 170 to 175.

Reviewer 17: Lines 262-263: “ These cultivars, bearing the 'BRS' prefix, are categorized into three distinct compatibility groups and exhibit diverse maturation cycles, including early, intermediate, and late maturation stages”. There is no information on the maturity group in Table 1. The superior genotypes identified in the study belong to which group?

Authors: Since these are other evaluations not related to the focus of this study, we refer only to some characteristics of the cultivars used as controls, which are distinguished by their better characterization. We have made sure to include the references associated with the characterization of these cultivars. Page 11, Lines 362 to 364.

Reviewer 18:  The paper deserves the table with the actual data of 68 genotypes performance for the studied traits in the supplement or in the paper body for the superior quality genotypes.

Authors: The table with the physicochemical characteristics of the 19 superior genotypes selected by various selection indices has been added on Page 9. To improve the understanding of the results and discussions, text has been included on Page 9, Lines 301 and 302; and Page 10, Lines 303 to 309. We have also prepared a table with information on all the evaluated genotypes to be presented as supplementary material.

Reviewer 1: The paper title does not reflect its essence. The authors did not study “genetic”  diversity and there is no information on the genes. This is phenotypic diversity for physicochemical traits using some genetic parameters. “…the most cultivated … genotypes…”. Is it likely that all 68 genotypes are “most cultivated”. Perhaps commonly grown.

Authors: As the term encompasses both genotypic effects and permanent and transient environmental effects, we decided to retain "genetic diversity" in the title. However, in response to the comment, we also agree that the title can be simplified: Genetic Variability in the Physicochemical Characteristics of Cultivated Coffea canephora Genotypes. Table 1, Lines 2 and 3.

Reviewer 2 Report

Comments and Suggestions for Authors

The manuscript explores the genetic variability of physicochemical characteristics in 68 genotypes of Coffea canephora, utilizing a variety of data types and methodologies. However, the study focuses exclusively on genotypes cultivated in the Western Amazon. Although the study evaluates 68 genotypes, the diversity represented may not fully capture the entire genetic variability available in Coffea canephora.

Comments: I request the authors to conduct dendrogram and AMOVA analyses, as well as statistical fitness analysis. Testing for the significance of variance component effects should be performed using the restricted maximum likelihood (REML) method. Additionally, please rank the genotypes based on the evaluation.

Reference:
Ceasar SA, Ramakrishnan M, Vinod KK, Roch GV, Upadhyaya HD, Baker A, et al. (2020). Phenotypic responses of foxtail millet (Setaria italica) genotypes to phosphate supply under greenhouse and natural field conditions. PLoS ONE, 15(6): e0233896. https://doi.org/10.1371/journal.pone.0233896

Comments on the Quality of English Language

 Minor editing of English language required.

Author Response

Letter to the Editor

We are resubmitting the manuscript entitled “Exploring the Genetic Variability of the Physicochemical Characteristics of the Most Cultivated Coffea canephora Genotypes” for further review following a careful revision.

The authors recognize the challenges involved in producing high-quality scientific work suitable for publication in a journal renowned for its scientific rigor. We have meticulously addressed each of the comments and suggestions received.

We would like to express our gratitude for the detailed feedback provided. All comments have been thoroughly considered in the revised version of the manuscript. At the end of the manuscript, we have addressed each contribution individually.

Revised sections of the text have been highlighted in blue to facilitate the identification of changes.

Thank you for your consideration.

Sincerely,

The authors.

Reviewer 2: The manuscript explores the genetic variability of physicochemical characteristics in 68 genotypes of Coffea canephora, utilizing a variety of data types and methodologies. However, the study focuses exclusively on genotypes cultivated in the Western Amazon. Although the study evaluates 68 genotypes, the diversity represented may not fully capture the entire genetic variability available in Coffea canephora.

Authors: We appreciate the comments and agree that the genetic variability measured does not encompass the full range of variability in this coffee species. However, our motivation for studying these plants arises from the fact that, despite their extensive cultivation, the genotypes studied in this work are unknown in many aspects. Thus, we focused on evaluating widely commercialized genotypes available in the public domain over time. We are thankful for the observations provided. We also acknowledge that before revising this work, we had not fully addressed a key issue: the classification of genotypes based on their combined concentrations of protein, phenolic compounds, and total sugars. Taking this into account, we understand the reviewer's perspective on the more practical implications of this study.

Reviewer 2: I request the authors to conduct dendrogram and AMOVA analyses, as well as statistical fitness analysis. Testing for the significance of variance component effects should be performed using the restricted maximum likelihood (REML) method. Additionally, please rank the genotypes based on the evaluation.

Authors: Peer review remains one of the best methods for assessing the quality and originality of scientific publications. We are very grateful to all our reviewers. In response to this comment, we have decided not to reanalyze the data using a maximum likelihood and generalized mixed models, as our data are balanced; a condition under which both methodologies converge to the same result. Regarding exploratory cluster analysis, we grouped the genotypes using a dendrogram with Mahalanobis distance to produce a divergence matrix and the UPGMA algorithm to construct a tree. However, we found that the principal components associated with the projection of characteristics provided a better interpretation of our results. Thus, we have decided to retain the analysis as presented. We have included the reference below concerning the methodologies used:

Reference:
Ceasar SA, Ramakrishnan M, Vinod KK, Roch GV, Upadhyaya HD, Baker A, et al. (2020). Phenotypic responses of foxtail millet (Setaria italica) genotypes to phosphate supply under greenhouse and natural field conditions. PLoS ONE, 15(6): e0233896. https://doi.org/10.1371/journal.pone.0233896

Round 2

Reviewer 1 Report

Comments and Suggestions for Authors

The authors did good job in addressing the manuscript comments and deficiencies. It is now ready for publication after addressing self-citation issue.

Author Response

Letter to the Editor

We are resubmitting the manuscript entitled “Genetic variability in the physicochemical characteristics of cultivated Coffea canephora genotypes” for review, after careful review of self-citations. 3 self-citations were removed: 12. Ramalho, A.R.; Rocha, R.B.; Souza, F.F.; Veneziano, W.; Teixeira, A.L. Genetic Progress in Productivity of Coffee Benefited from the Selection of ‘Conilon’1 Coffee Clones. Revista Ciencia Agronomica 2016, 47, 516–523, doi:10.5935/1806-6690.20160062. 26. de Morais, J.A.; Rocha, R.B.; Alves, E.A.; Espindula, M.C.; Teixeira, A.L.; de Souza, C.A. Beverage Quality of Coffea Canephora Genotypes in the Western Amazon, Brazil. Acta Sci Agron 2021, 43, doi:10.4025/ACTASCIAGRON.V43I1.52095. 29. Rocha, R.B.; Teixeira, A.L.; Ramalho, A.R.; Espindula, M.C.; Lunz, A.M.P.; Souza, F. de F. Coffea Canephora Breeding: Estimated and Achieved Gains from Selection in the Western Amazon, Brazil. Ciência Rural 2021, 51, doi:10.1590/0103-8478cr20200713.

The inclusion of new citations are highlighted in blue, in the references, to facilitate the identification of changes. In total, the self-citation rate was 17.07% (7/41). Aware of the maximum rate of 15% recommended by the magazine, however, coffees grown in the Western Amazon are still studied by few researchers, where the articles cited on the subject tend to be collaborated by the authors who are part of the manuscript in question. Thank you for your consideration.

Sincerely,

The authors.

Reviewer 2 Report

Comments and Suggestions for Authors

The authors have addressed the comments; however, according to the COPE Guidelines, the self-citation rate should not exceed 20%. Therefore, I request the authors to reduce their self-citations.

Author Response

(The authors gave the same response as above.)
